# Predictive Modeling of Machining Temperatures with Force–Temperature Correlation Using Cutting Mechanics and Constitutive Relation

**DOI:** 10.3390/ma12020284

**Published:** 2019-01-16

**Authors:** Jinqiang Ning, Steven Y. Liang

**Affiliations:** George W. Woodruff School of Mechanical Engineering, Georgia Institute of Technology, 801 Ferst Drive, Atlanta, GA 30332-0405, USA

**Keywords:** machining temperatures at two deformation zones, force–temperature correlation through analytical modeling, high computational efficiency, real-time prediction

## Abstract

Elevated temperature in the machining process is detrimental to cutting tools—a result of the effect of thermal softening and material diffusion. Material diffusion also deteriorates the quality of the machined part. Measuring or predicting machining temperatures is important for the optimization of the machining process, but experimental temperature measurement is difficult and inconvenient because of the complex contact phenomena between tools and workpieces, and because of restricted accessibility during the machining process. This paper presents an original analytical model for fast prediction of machining temperatures at two deformation zones in orthogonal cutting, namely the primary shear zone and the tool–chip interface. Temperatures were predicted based on a correlation between force and temperature using the mechanics of the cutting process and material constitutive relation. Minimization of the differences between calculated material flow stresses using a mechanics model and a constitutive model yielded an estimate of machining temperatures. Experimental forces, cutting condition parameters, and constitutive model constants were inputs, while machining forces were easily measurable by a piezoelectric dynamometer. Machining temperatures of AISI 1045 steel were predicted under various cutting conditions to demonstrate the predictive capability of each presented model. Close agreements were observed by verifying them against documented values in the literature. The influence of model inputs and computational efficiency were further investigated. The presented model has high computational efficiency that allows real-time prediction and low experimental complexity, considering the easily measurable input variables.

## 1. Introduction 

Machining is one of the most widely used manufacturing processes because of its fast speed and applicability to a broad class of materials. Karpat et al. studied the machining process of steel and aluminum alloys [1]. Danish et al. studied the machining process of magnesium alloy under dry and cryogenic cutting conditions [2]. Ning et al. studied the machining process of ultra-fine-grained titanium [3]. Machining temperature has a significant influence on tool performance and the quality of a machining part as a result of the softening effect and diffusion. Coolant [4], laser power [5], and magnetic flux [6] have been utilized to effectively control temperature in the machining process. The capabilities of temperature measurement and prediction are critical for optimizing the machining process. Predicted temperatures can be utilized to further explore machining forces, tool wear, material diffusion, etc.

Different experimental methods have been used to measure machining temperatures. Embedded thermocouples [7], tool–work thermocouples [8], infrared photography and pyrometers [9], a metallographic technique based on microstructure and hardness [10], and a metallographic technique using powders with known melting temperature [11] have been utilized in the past to measure machining temperatures. Unfortunately, experimental measurement is difficult and inconvenient to record due to the complex contact phenomena between cutting tools and workpieces, and because of restricted accessibility during the machining process [12].

Numerical methods using finite element (FE) analysis and analytical methods were developed to predict temperatures, and numerical methods using FE models have made considerable progress in their ability to predict machining processes, including machining forces, temperatures, residual stress, and chip morphology. Umbrello et al. developed an FE model to predict conventional high-speed machining processes [13]. Liu et al. developed another FE model to predict sequential machining processes [14]. Yen et al. investigated the influence of tool geometry on machining prediction using an FE model to optimize tool edge design [15]. Özel et al. investigated the influence of tool coating on machining prediction using an FE model to demonstrate the advantages of coated tool design [16]. Umbrello et al. demonstrated that machining prediction using an FE model was very sensitive to the materials constitutive model constants [17]. Arrazola et al. demonstrated that consideration of the friction coefficient at the tool–chip-work interface results in improved accuracy of machining prediction using an FE model [18]. Unfortunately, the high computational cost and low computational efficiency of numerical methods have been major limitations preventing real-time prediction and optimization with process-parameter planning.

To overcome these limitations, analytical methods were developed that could predict machining processes with comparable accuracy along with considerably high computational efficiency [19,20]. The chip formation model was modified primarily to predict machining forces in orthogonal cutting, in which the Johnson–Cook constitutive model (J–C model) is employed to calculate material flow stress. Temperatures in the J–C model, specifically at the primary shear zone (PSZ) and tool–chip interface (alternatively named secondary shear zone or SSZ), are calculated using heat partition equations as intermediate variables for force prediction [21]. Temperatures at the PSZ can also be explicitly determined by observing the energy balance between plastic works caused by shear deformation and generated heat [22]. Komanduri et al. developed a temperature model that used two heat sources at the PSZ and SSZ to predict temperature distribution at the chip formation zone [23]. The heat source caused by shear deformation at the PSZ was observed using a moving heat source solution with boundary conditions defined by appropriate image sources. The heat source caused by the friction between the tool and chip at the SSZ was observed by comparing the equivalence between two heat source solutions, namely a moving heat source in the chip and a stationary heat source in the tool. This model was further developed by considering the thermal properties of tools and tool-wear under oblique cutting conditions [24,25]. Improved prediction accuracy was reported after results were validated against experimental measurements. Shalaby et al. developed a temperature model to predict machining temperatures by considering shear deformation and friction at two precision-turning deformations zones [26]. However, these developed analytical models need temperature-dependent material properties of the workpiece that must be obtained from extensive material property tests, which are inconvenient. The shear angle and strain rate constants in the chip formation model are determined iteratively with complex mathematical calculations, which limits optimal computational efficiency, and thus restricts real-time temperature prediction.

In this work, the machining temperatures at two deformation zones were predicted by an original temperature model using the correlation between machining forces and temperatures. Machining forces can be easily and reliably measured using a piezoelectric dynamometer as reported in the literature [27]. The temperatures were correlated to forces using a constitutive model and a mechanics model with stress calculations at the PSZ and SSZ, respectively. AISI 1045 steel was chosen to test the presented models under various cutting conditions. The predicted temperatures were validated against documented values in the literature [21,28]. For comparison, the analytical model reported in the previous work used the chip-thickness and constant-material-flow-rate assumption that prevents real-time temperature prediction and optimized prediction accuracy [29]. More details of the previous model and its predictive capability can be found in reference [30]. The experimental techniques and developed models used to investigate the machining process are summarized in Table 1. In addition, sensitivity analyses were conducted to investigate the influence of input forces and J–C model constants on prediction accuracy.

## 2. Methodology 

Machining temperatures were predicted at the PSZ and SSZ in orthogonal cutting with machining forces used as inputs. Machining temperatures and forces were correlated using the mechanics of the cutting process and material constitutive relation. An orthogonal cutting configuration is illustrated as in Figure 1, where $F_{c}\;\mathrm{and}\;F_{t}$ are the cutting force and thrust force, respectively, that can be measured using a piezoelectric dynamometer. α,β, and ϕ are the tool–rake angle, friction angle, and shear angle, respectively. $V_{c},V_{s},\;\mathrm{and}\;V$ are the chip velocity, shear velocity, and cutting velocity, respectively. w is the cutting width that is not shown. Steady-state condition and plane-strain condition were enforced in the temperature prediction.

The stresses at two shear zones were calculated using the mechanics of the orthogonal cutting process with the given cutting force and thrust force, (see the Appendix A, Table A1, Table A2). The shear stresses at the PSZ $\left(k_{AB}\right)$ and SSZ $\left(\tau_{int}\right)$ can be expressed using a mechanics model as
(1)$$k_{AB}=\;\frac{F_{s}}{l_{AB}w}$$
(2)$$\tau_{int}=\;\frac{F}{hw}$$
where $F_{s}$ and F can be calculated with determined by the rake angle (α), friction angle (β), shear angle (ϕ), and experimental forces ($F_{c}$, $F_{t}$). $l_{AB}$ and h are the length of the PSZ and SSZ, respectively.

The friction angle (β) can be calculated from the force circle as
(3)$$\beta -\alpha =\mathrm{atan}\left(\frac{F_{t}}{F_{c}}\right)$$

The shear angle was determined by minimizing the cutting work according to the shear angle solution presented by Ernst and Merchant [31]. The cutting work was proportional to the cutting force, which can be expressed as
(4)$$F_{c}=\;\tau \frac{wt_{1}}{\sin\phi }\frac{\cos\left(\beta -\alpha \right)}{\cos\left(\phi +\beta -\alpha \right)}$$
where τ is the shear stress, w is the width of cutting, $\mathrm{and}\;t_{1}$ is the depth of cutting. 

The shear angle (ϕ) can then be expressed by differentiating the above equation as
(5)$$\phi =\;\frac{\pi }{4}-\frac{\beta }{2}+\frac{\alpha }{2}$$

The angle between shear force and resultant force (θ) can be calculated from the force circle as
(6)θ= ϕ+β+α

The lengths of the PSZ $\left(l_{AB}\right)$ and SSZ (h) can be expressed as
(7)$$l_{AB}=\frac{t_{1}}{sin\phi }$$
(8)$$h=\;\frac{t_{1}sin\theta }{cos\lambda sin\phi }\left(1+\frac{C_{0}n_{eq}}{3\left(1+2\left(\frac{\pi }{4}\;-\;\phi \right)-C_{0}n_{eq}\right)}\right)$$

The stresses at the two shear zones can also be calculated using the constitutive relation. The Johnson–Cook constitutive model (J–C model) was chosen for the calculation with consideration of the strain hardening effect, the strain-rate hardening effect, and the thermal softening effect. The J–C model can be expressed as
(9)$$\sigma =\left(A+B\varepsilon^{n}\right)\left[1+C\ln\left(\frac{\dot{\varepsilon }}{\dot{\varepsilon_{0}}}\right)\right]\left[1-{\left(\frac{T-\;T_{r}}{T_{m}-\;T_{r}}\right)}^{m}\right]$$
where *A*, *B*, *C*, *m*, and *n* are five material constants that can be determined by various approaches such as Split­–Hopkinson Pressure Bar (SHPB) tests [32], numerical methods [33], and analytical methods [34]. The analytical methods have less experimental complexity and high computational efficiency compared to the experimental tests and numerical methods, respectively, as discussed in the literature [35,36].

The shear stresses at the PSZ $\left(k_{AB}^{\prime }\right)$ and SSZ $\left(k_{int}\right)$ can be calculated using the J–C model with the von Mises yield criterion as
(10)$$k_{AB}^{\prime }=\;\frac{\sigma_{AB}}{\sqrt{3}}\;=\frac{1}{\sqrt{3}}\left(A+B\varepsilon_{AB}^{n}\right)\left(1+Cln\frac{\dot{\varepsilon_{AB}}}{\dot{\varepsilon_{0}}}\right)\left(1-{\left(\frac{T_{AB}\;-\;T_{r}}{T_{m}\;-T_{r}}\right)}^{m}\right)$$
(11)$$k_{int}=\frac{1}{\sqrt{3}}\left(A+B\varepsilon_{int}^{n}\right)\left(1+C\ln\frac{\dot{\varepsilon_{int}}}{\dot{\varepsilon_{0}}}\right)\left(1-{\left(\frac{T_{int}\;-\;T_{r}}{T_{m}\;-T_{r}}\right)}^{m}\right)$$
where strains and strain rates are calculated as
(12)$$\varepsilon_{AB}=\;\frac{\gamma_{AB}}{\sqrt{3}}=\;\frac{cos\alpha }{2\sqrt{3}sin\phi \cos\left(\phi -\alpha \right)}$$
(13)$$\dot{\varepsilon_{AB}}=\;\frac{\dot{\gamma_{AB}}}{\sqrt{3}}=\;C_{0}\frac{V_{s}}{\sqrt{3}l_{AB}}$$
(14)$$\varepsilon_{int}=\frac{\gamma_{int}}{\sqrt{3}}=2\varepsilon_{AB}+\frac{h}{2\sqrt{3}\delta t_{2}}$$
(15)$$\dot{\varepsilon_{int}}=\frac{\dot{\gamma_{int}}}{\sqrt{3}}=\frac{V_{c}}{\sqrt{3}\delta t_{2}}$$

The temperatures were determined by minimizing the difference between the stress calculated using the mechanics model and the same stress calculated using the J–C model at each shear zone as illustrated in Figure 2. Iterations with a defined temperature range, specifically a range between room temperature ($T_{r}$) and material melting temperature ($T_{m}$), were used in the minimization for temperature prediction. The cutting condition parameters, J–C model constants, and experimental forces were given as the inputs, and the average temperatures at the PSZ and SSZ were calculated as the outputs.

The machining temperature was predicted in the presented model using the correlation between the forces and temperature with the given forces as inputs, permitting it less mathematical complexity and thus higher computational efficiency compared to the chip formation model. The presented model also had less experimental complexity because of the following: (1) Experimental forces were reliable and easily measurable using a three-axial piezoelectric dynamometer. (2) Temperature-sensitive material properties such as thermal conductivity and specific heat (which require extensive material property tests to be obtained) were not needed in the presented model. In addition, the high computational efficiency allowed real-time temperature prediction with real-time force data. However, there were some limitations of the presented model: (1) The presented model only predicted the average temperatures at the PSZ and SSZ. (2) Prediction accuracy relied on accurate model inputs, such as forces and J–C model constants. 

To further investigate the advantages and disadvantages of the presented model, machining temperatures were predicted in the orthogonal cutting of AISI 1045 steel under various cutting conditions. The following tasks were performed: (1) An investigation of prediction accuracy was conducted by validating against documented values in the literature. (2) An investigation was conducted into computational efficiency in terms of computational time. (3) An investigation was conducted on the influence of input machining forces on prediction accuracy. (4) An investigation was conducted on the influence of multiple sets of available J–C model constants on prediction accuracy. (5) A discussion was carried out on the usefulness of the predicted temperature data and future works.

## 3. Results and Discussion 

In this work, machining temperatures were predicted in the orthogonal cutting of AISI 1045 steel under various cutting conditions. The model inputs of cutting condition parameters and machining forces were adopted from the literature [21,28] as presented in Table 2. The documented values in tests 1–4 were calculated using an improved chip formation model, in which machining temperatures were calculated with heat partition equations at two shear zones. The documented values in tests 5–8 were calculated using an extended chip formation model, in which machining temperatures were calculated based on two heat sources at the PSZ and SSZ. The J–C model constants of AISI 1045 steel were adopted from the literature [32], in which SHPB tests were conducted. 

In the presented model, the temperature at the PSZ ($T_{AB}$) was determined by minimizing the difference between the stress ($k_{AB}$) and the stress ($k_{AB}^{\prime }$), while the temperature at the SSZ ($T_{int}$) was determined by minimizing the difference between the stress ($\tau_{int}$) and the stress ($k_{int}$). The predicted temperatures were validated by the documented values in the literature as presented in Table 3. The documented values were validated through force comparison against experimental measurements using a three-axial piezoelectric dynamometer in orthogonal cutting tests [27] (the temperatures are intermediate variables in calculating machining forces). Good agreements were observed upon force validation. Other calculated variables of the shear angle and stresses are shown in Table 4. The temperature prediction was carried out using a MATLAB program on a personal computer running at 2.8 GHz. To investigate the computational efficiency, the computational time for each prediction was recorded as shown in Table 3. The average computational time was 0.27 s, which allowed real-time temperature prediction during the machining process and cutting-parameter planning with a trial-and-error method.

Good agreements were observed between the predicted temperatures and documented values as shown in Figure 3. The predicted temperatures at the PSZ and SSZ were generally larger than the documented values because of the assumption of a perfectly sharp cutting edge in the chip formation model; underestimated machining forces and temperatures were reported with this assumption in the literature, which affected the literature’s predictions and experimental measurements [21,37]. The deviations between predicted temperatures and documented values might also have been affected by the deviation of input machining forces due to vibrations, which were frequently observed in heavy-duty operations [38] and in machining difficult-to-cut materials [39]. A study of the influence of input forces on the accuracy of temperature prediction is needed.

To investigate the influence of input experimental forces on the predicted temperatures, the input cutting force and thrust force were deliberately changed (separately) up to ±20% from their original values under the test 1 cutting condition. The prediction error was calculated by comparing to the documented values from the literature [28], as shown in Figure 4. For the temperature at the PSZ, the prediction error using input forces was found at the global minima. For the temperature at the SSZ, the prediction error using input forces was found near the local minima, with relatively larger values. The temperature prediction at the PSZ was more sensitive than the temperature at the SSZ in the orthogonal machining of AISI 1045 steel. The temperature deviations at the PSZ were much larger than that at the SSZ with the same amount of input-force deviations. In addition, the predicted temperatures at the PSZ were more sensitive to input-cutting-force deviations. 

Multiple sets of J–C constants are available for AISI 1045 steel (shown in Table 5). They were determined using different methods. To investigate the influence of J–C model constants on the accuracy of temperature prediction, different sets of J–C constants were used for prediction as shown in Table 5. The predicted temperatures were validated against documented values under the test 7 cutting condition as illustrated in Figure 5. Acceptable agreements were observed upon temperature validation. 

The predicted temperatures at the PSZ and SSZ are sufficient for the further investigation of machining forces [1], tool wear [43], and material diffusion [44], as reported in the literature. High computational efficiency allows process-parameter planning with a trial-and-error calculation to determine the desired temperature conditions. AISI 1045 steel was chosen for this study because of the ready availability of machining data and J–C model constants. Other metal materials should be investigated to extend the applicability of the presented model in future works.

## 4. Conclusions

This work presents an original analytical model for temperature prediction in the machining process. Machining temperatures and forces were correlated with the mechanics of the cutting process and constitutive relation. The stresses at two shear zones were calculated with the input cutting force and thrust force using a mechanics model. The same stresses were also calculated with unknown temperatures using the J–C model. The minimization between calculated stresses yielded an estimation of unknown temperatures at the PSZ and SSZ. Good agreements were observed based upon validation against the documented temperatures in the literature. The presented model improved an understanding of the force–temperature relationship in the machining process using mathematical calculation. The influence of input-force deviations and J–C model constants on the accuracy of temperature prediction was investigated with sensitivity analyses. Temperature prediction at the PSZ was more susceptible to input-force deviations than temperature prediction at the SSZ. Acceptable predicted accuracy was achieved with multiple sets of available J–C constants. In addition, the average computational time for temperature prediction using the presented model was about 0.27 s, which allowed for real-time temperature prediction and process-parameter planning using trial-and-error calculations. Having achieved a high level of prediction accuracy, high computational efficiency, and low experimental complexity, this presented temperature model can be employed in the future for investigating temperature in the machining process. The applicability of the presented temperature model can further be used for prediction in machining different metals.

## Figures and Tables

**Figure 1 materials-12-00284-f001:**
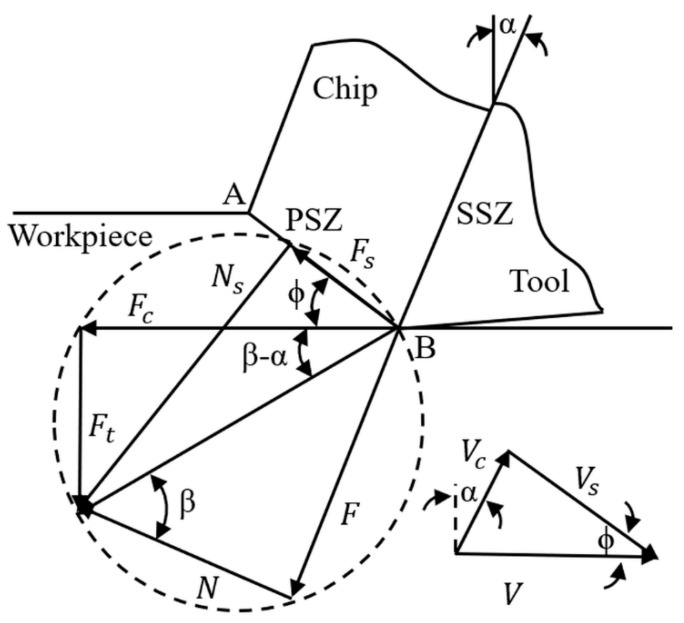
Schematic drawing of the orthogonal cutting process using a force circle. PSZ and SSZ denote the primary shear zone and secondary shear zone, respectively.

**Figure 2 materials-12-00284-f002:**
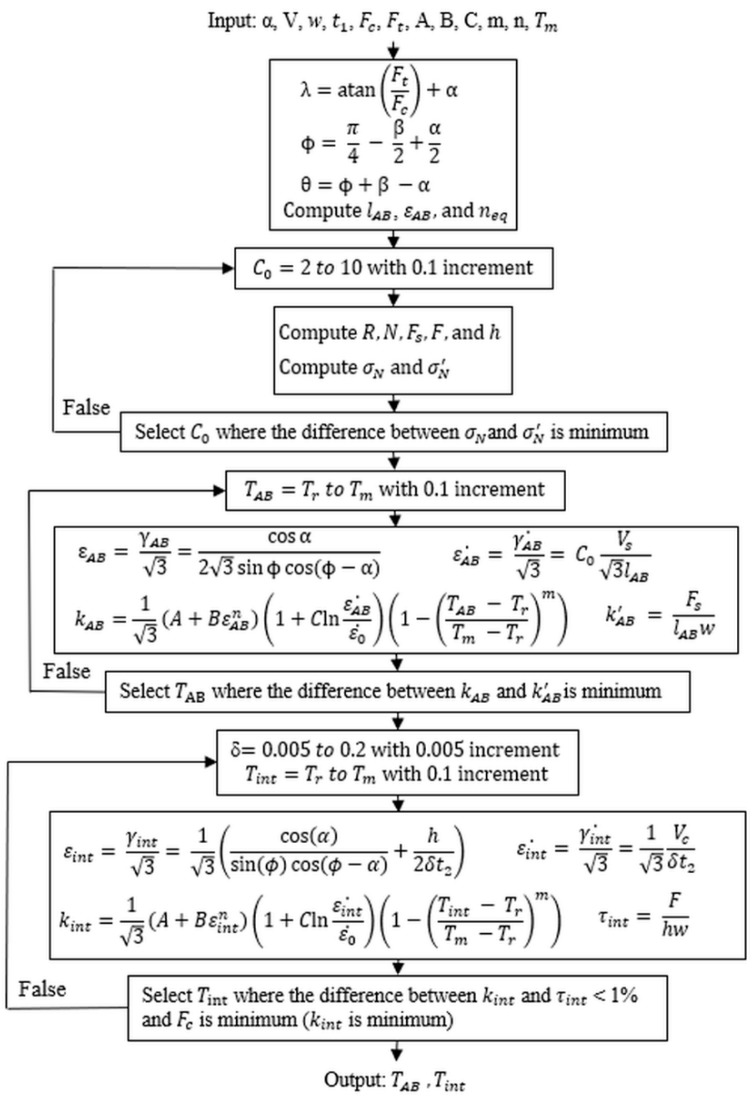
The algorithm of temperature predictions in the presented model.

**Figure 3 materials-12-00284-f003:**
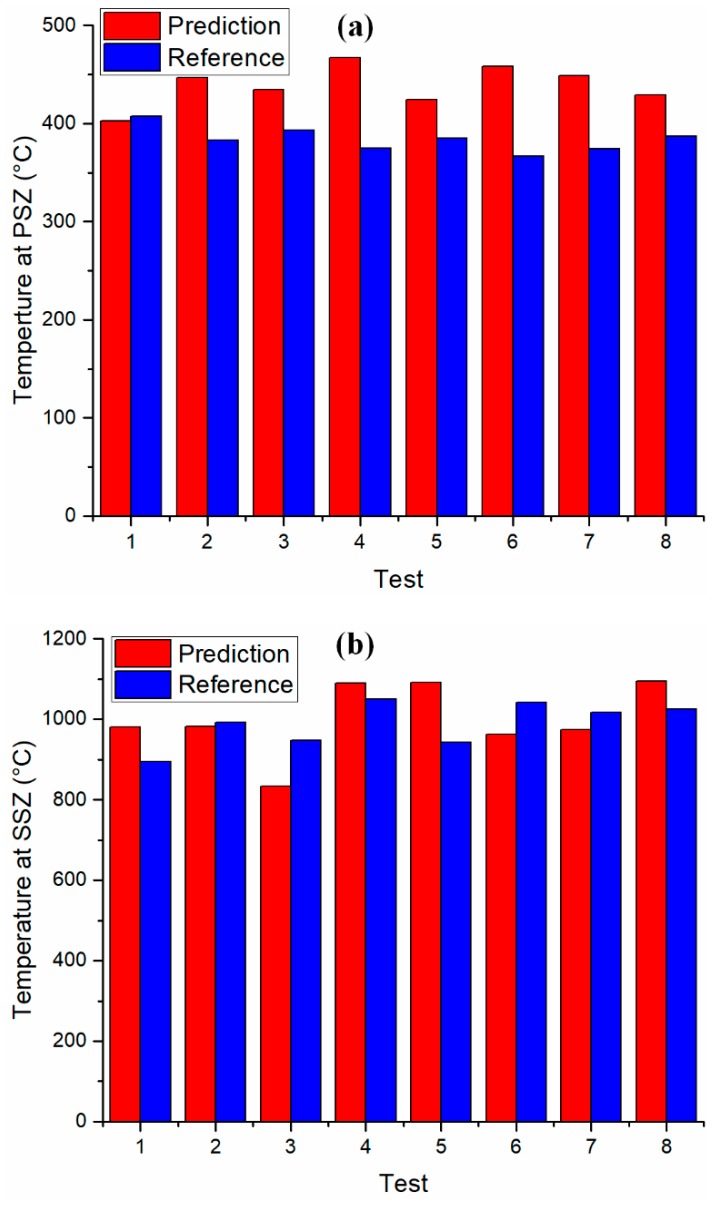
Temperature validation against documented values in the orthogonal cutting of AISI 1045 steel [21,28]. (**a**) Validation of temperature prediction at the primary shear zone. (**b**) Validation of temperature prediction at the secondary shear zone.

**Figure 4 materials-12-00284-f004:**
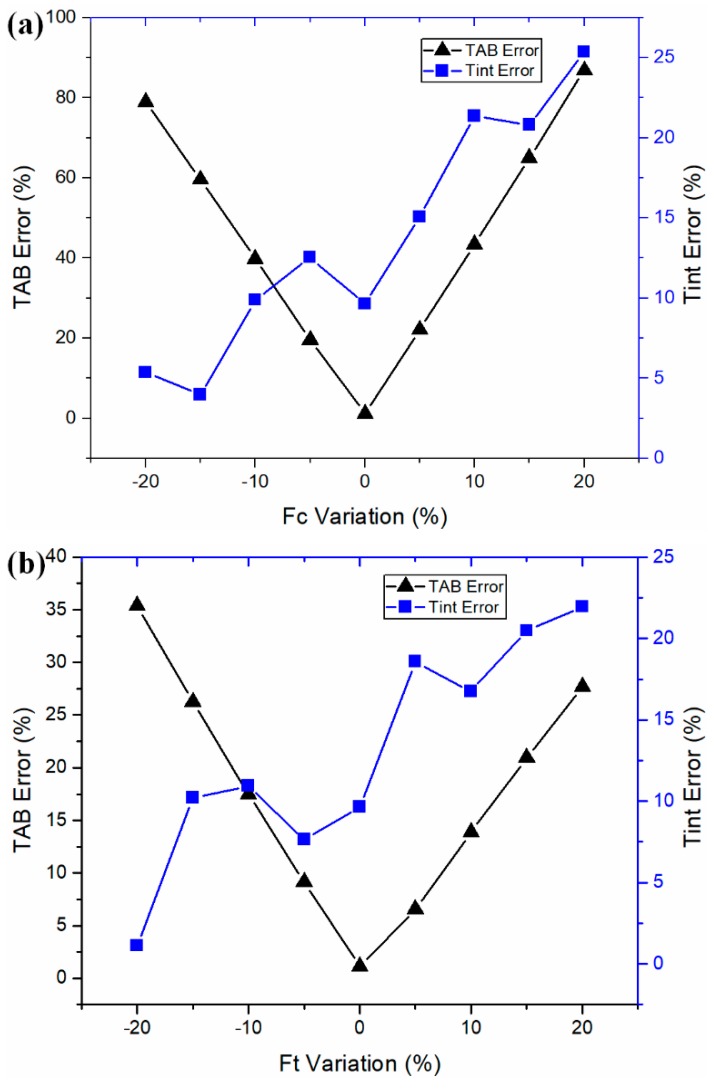
Sensitivity analyses of (**a**) cutting force and (**b**) thrust force on the temperature prediction.

**Figure 5 materials-12-00284-f005:**
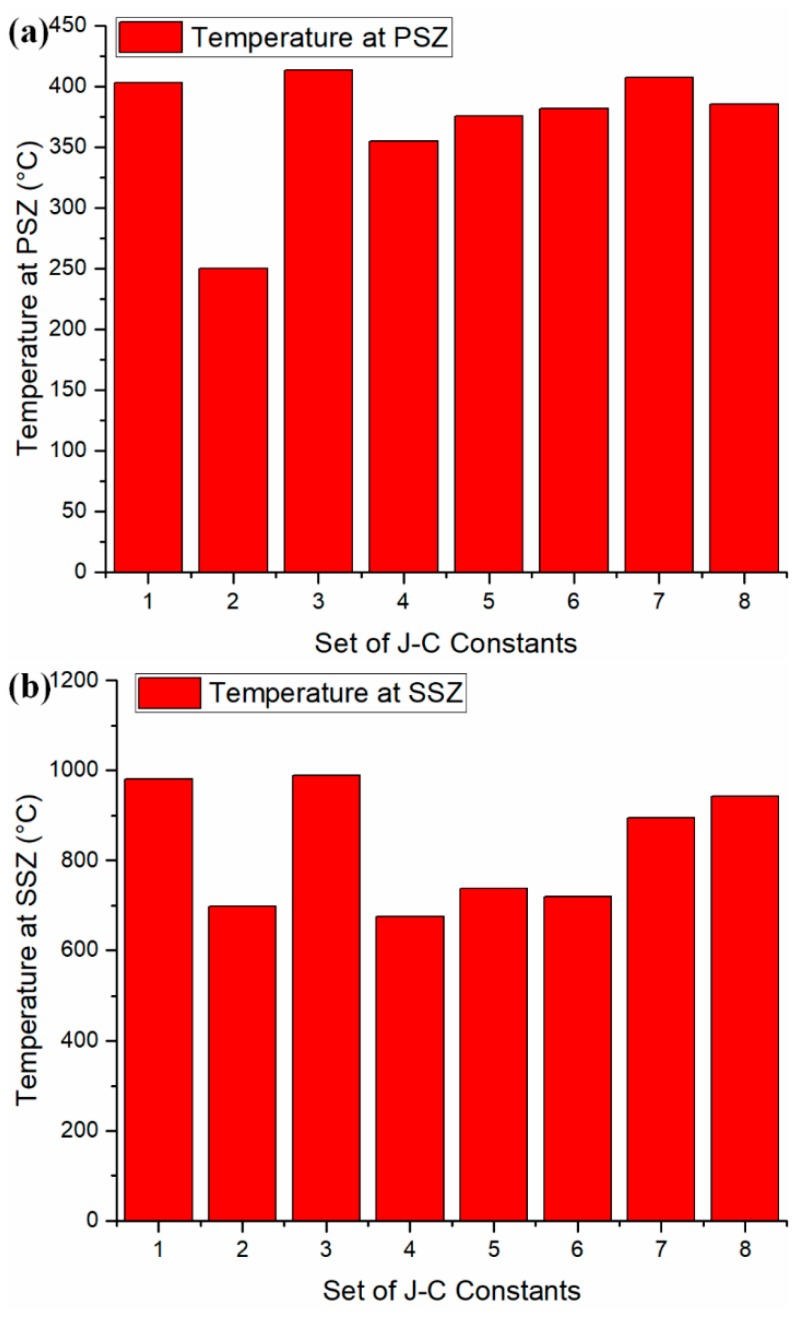
Sensitivity analysis of Johnson–Cook model constants on temperature predictions at (**a**) the primary shear zone; and (**b**) the secondary shear zone. The predicted temperatures under sets 1–6 were predicted using adopted J–C constants that were determined using different methodologies. The temperatures under sets 7 and 8 were adopted from the literature [21,28].

**Table 1 materials-12-00284-t001:** Summary of experimental and modeling methods in the investigation of the machining process.

Methods	Experimental Techniques	Numerical Methods	Analytical Methods
	Embedded thermocouple [7], tool–work thermocouple [8], infrared technique [9], graphic techniques [10,11]	FEA for machining forces, temperature distribution, residual stress, and chip morphology [13,14]	Chip formation model [21], Komanduri’s model [23], Shalaby’s model [26], Ning’s model [29]
Major advantage	Sufficient accuracy for in-situ/ post-processing measurement	Sufficient prediction capability	High computational efficiency
Major disadvantage	High experimental complexity	High computational cost	Complex input requirement; high mathematical complexity

**Table 2 materials-12-00284-t002:** Cutting condition parameters in the orthogonal machining of AISI 1045 steel (*w* = 2 mm, *α* = −7°, $T_{0\;}$ = 25 °C) [21,28].

Test	V (m/min)	$t_{1}\;(\mathrm{mm})$	$F_{cR}(N)$	$F_{tR}(N)$	$T_{ABR}\;({}^{\circ}C)$	$T_{intR}\;({}^{\circ}C)$
1	200	0.15	625.42	439.86	407.39	895.07
2	200	0.3	1077.7	637.19	383.1	992.44
3	300	0.15	574.55	364.74	393.31	947.81
4	300	0.3	1003.6	531.84	374.64	1049.8
5	200	0.15	576	500	385	942
6	200	0.3	1007	740	367	1042
7	300	0.15	533	478	374	1017
8	300	0.3	1041	628	387	1025

Note: Temperature and force values in Tests 1–4 were adopted from the literature [28] using an improved chip formation model, and Tests 6–8 were adopted from the literature [21] using an extended chip formation model. Subscript *R* denotes documented values.

**Table 3 materials-12-00284-t003:** Temperature prediction and validation in the orthogonal machining of AISI 1045 steel.

Test	$T_{AB}\;({}^{\circ}C)$	$T_{int}\;({}^{\circ}C)$	$T_{AB}\;\mathrm{Deviation}\;(\%)$	$T_{int}\;\mathrm{Deviation}\;(\%)$	t (s)
1	402.81	981.33	1.12	9.64	0.389
2	446.86	982.63	16.64	0.99	0.252
3	434.20	834.04	10.40	12.00	0.258
4	467.29	1089.66	24.73	3.80	0.243
5	424.04	1091.96	10.14	15.92	0.293
6	458.62	962.50	24.96	7.63	0.239
7	448.55	974.66	19.93	4.16	0.240
8	428.82	1094.65	10.81	6.79	0.242

Note: $T_{AB}$ and $T_{int}$ denote the average temperatures at the PSZ and SSZ respectively.

**Table 4 materials-12-00284-t004:** Calculated shear angle and stresses at the PSZ and SSZ.

Test	ϕ (degs)	$k_{AB}\;(\mathrm{MPa})$	$k_{AB}^{\prime }\;(\mathrm{MPa})$	$\tau_{int}\;(\mathrm{MPa})$	$k_{int}\;(\mathrm{MPa})$
1	27.44	541.25	541.37	455.90	455.90
2	29.70	496.39	496.50	400.62	400.62
3	28.80	512.32	512.43	358.80	358.80
4	31.04	480.25	480.39	330.97	330.97
5	28.85	528.96	529.10	397.45	397.45
6	31.05	490.90	491.01	361.49	361.49
7	30.23	505.30	505.42	311.87	311.86
8	31.08	478.63	478.75	296.56	296.56

**Table 5 materials-12-00284-t005:** Johnson–Cook constitutive model constants of AISI 1045 steel ($T_{m}$ = 1460 °C; $\dot{\varepsilon_{0}}=1$ ).

Set	Method	*A* (MPa)	*B* (MPa)	C	m	n
1	SHPB [32]	553.1	600.8	0.0134	1	0.234
2	FEA [40]	546	487	0.03	0.672	0.25
3	Analytical Modeling [41]	451.6	819.5	0.0000009	1.0955	0.1736
4	PSO [42]	646.19	517.7	0.0102	0.94054	0.24597
5	PSO-c [42]	731.63	518.7	0.00571	0.94054	0.3241
6	CPSO [42]	546.83	609.35	0.01376	0.94053	0.2127

Note: SHPB: Split–Hopkinson pressure bar; PSO: particle swarm optimization algorithm; CPSO: corporative particle swarm optimization algorithm.

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
