# Peer review of "Predictive Modeling of Machining Temperatures with Force–Temperature Correlation Using Cutting Mechanics and Constitutive Relation"

_materials, 2019, doi:10.3390/ma12020284_

Round 1

Reviewer 1 Report

Paper title:

 Predictive Modeling of Machining Temperatures with Force-Temperature Correlation using cutting mechanics and constitutive relation

The coauthors presented a mathematical model to predict the cutting temperatures based on Merchant and Johnson-Cook models. They have compared the predicted values of temperatures with previously published results.

Careful reading of this paper shows that the methodology used to predict the cutting (machining) temperatures is not clear and hard follow. Moreover, the workpiece and cutting tool material thermo-physical properties (thermal conductivity, volumetric heat capacity) and their variation with the different temperatures have not been considered in the calculations. This would lead to a high degree of uncertainty in the presented results.   

The reviewer comments are the following points:

1-      Nomenclatures section has to be added to the manuscript.

2-      Line 11, "thermal softening effect", please clarify that this effect is detrimental to the tool material. On the other hand, this effect may have a positive impact on the machining operation due to the consequent decrease of the cutting forces.

3-      Line 15, "at the two shear zones", this is not correct. Just one shear zone exists during metal cutting. This sentence can be replaced by "at the two deformation zones".

4-      Line 100, it is not clear why the shear angle is determined by minimizing the cutting work.

5-      How did the coauthors predict the strain and strain rate in J-C equations?

6-      How did the coauthors predict equation 10, 11?

7-      The velocity triangle in Figure 1 is not correct, the cutting speed (V) should be the resultant.

8-      The coauthors claimed in line 73 that they predicted the machining temperature, this cannot be seen in the results. They have predicted the temperatures at the two deformation zones.

9-      Line 115, "The temperatures are determined by minimizing the difference between stress calculated using mechanic model and the same stress calculated using J-C model at each shear zone as illustrated as in Figure 2", this is not understood. What is the basis to perform this minimization?   

10-  Figure 3, 5, the temperatures in the secondary deformation zone are more than in the primary zone, this is not correct. The work consumed to overcome the shearing action at the primary deformation zone is greater than the work consumed to overcome the friction at the secondary defamation zone, however the temperature at the secondary zone should be greater. Please refer to DOI: 10.1007/s00170-018-2389-8

Author Response

The authors appreciate the reviewer’s efforts on improving our manuscript. A detailed response to the reviewer’s comments is attached. 

Reviewer 2 Report

See attached

Author Response

(The authors gave the same response as above.)

Round 2

Reviewer 1 Report

The coauthors have addressed the comments.

Author Response

The authors would like to express their sincere appreciations for reviewer’s time and effort on improving our manuscript.

Reviewer 2 Report

all answers have been met, paper can now be accepted

Author Response

(The authors gave the same response as above.)
